# Impact of Dietary Fats on Cardiovascular Disease with a Specific Focus on Omega-3 Fatty Acids

**DOI:** 10.3390/jcm11226652

**Published:** 2022-11-09

**Authors:** Aldo Pietro Maggioni, Giuseppe Poli, Pier Mannuccio Mannucci

**Affiliations:** 1ANMCO Research Center, Heart Care Foundation, 50121 Florence, Italy; 2Department of Clinical and Biological Sciences, University of Turin, San Luigi Hospital, 10043 Turin, Italy; 3Angelo Bianchi Bonomi Hemophilia and Thrombosis Center, IRCCS Foundation Ca’ Granda Maggiore Policlinico Hospital, 20122 Milan, Italy

**Keywords:** dietary fats, fat oxidation, nutritional research, n-3 PUFA, cardiovascular prevention

## Abstract

Dietary habits have major implications as causes of death globally, particularly in terms of cardiovascular disease, cancer and diabetes, but to precisely define the role of the single components of diet in terms of cardiovascular risk is not an easy task, since current epidemiological cohorts do not include sufficient information regarding all the confounding factors typical of nutritional associations. As an example, complex and multifactorial are the possible nutritional or detrimental effects of dietary fats, due to the huge variety of lipid metabolites originating from either the enzymatic or non-enzymatic oxidation of polyunsaturated fatty acids, cholesterol and phospholipids. The area of research that has allowed the benefit/risk profile of a dietary supplement to be tested with controlled studies is that of omega-3 fatty acids. Omega-3 fatty acids have showed a potential therapeutic role only in secondary cardiovascular prevention, while controlled studies in primary prevention have consistently produced neutral results. Despite some favorable evidence in patients with chronic heart failure; a treatment with n-3 PUFA in this clinical context is presently overlooked. The potential risk of atrial fibrillation, especially when n-3 PUFA are used in high doses, is still under scrutiny.

## 1. Introduction

Browsing the literature on the epidemiology of food components means becoming lost in an infinite sea. In the last 3 years, more than 200,000 peer-reviewed publications have appeared on these topics (PubMed Research June 2022). In recent meta-analyses of prospective cohort studies, nearly all foods analyzed revealed statistically significant associations with mortality risk, and among those, dietary fats certainly deserve high consideration. For this reason, we aimed to take into consideration a limited number of recent papers, with the goal to try to answer a short set of practical questions.

## 2. Which Foods, or Food Shortages, Can Impact Mortality in Different AREAS of the World? What Kind of Non-Communicable Disease Can Be Impacted by Different Dietary Habits?

A very comprehensive publication discussing this topic has been recently published [1]. A systematic analysis from the Global Burden of Disease study, an initiative sponsored by the Melinda and Bill Gates Foundation, evaluated in 2017 across 195 different countries the role of the consumption of foods and nutrients or their suboptimal intake on the mortality and morbidity for non-communicable diseases. In 2017, cardiovascular disease was the leading cause of food-related deaths globally (10 million deaths), followed by cancer (913,090 deaths) and type 2 diabetes (338,714 deaths) (Figure 1). Although the impact of individual dietary factors varied between countries, the suboptimal intake of three dietary elements (low intake of whole grains and fruits, high intake of sodium) accounted for more than 50% of diet-attributable deaths (Figure 1).

Therefore, although sodium, sugar and fat have been the main focus of dietary policy recommendations for the past two decades, the study demonstrates that the main dietary risk factors for mortality are diets high in sodium, low in whole grains, low in fruit, nuts and seeds, low in vegetables and low in omega-3 fatty acids. These data suggest that dietary policies that focus on promoting the intake of dietary components for which the actual intake is under the optimal level may have a greater effect than policies targeting only sugar and fat reduction. Given the complexity of eating behaviors, diet improvement requires the active collaboration of a variety of actors across the food system, along with policies targeting multiple sectors of the food system [1].

## 3. Is It True That the Consumption of Saturated Fat Has a Negative Impact on Cardiovascular Mortality?

Recently, a review on the state of the art on the role of saturated fat intake on health profiles was published [2]. This article also allowed a reassessment of the current guideline recommendations regarding food intake and suggested some proposals for their modification. The authors emphasized that the US dietary guidelines [3] recommend limiting the intake of saturated fatty acid (SFA) to <10% of total calories in order to reduce cardiovascular disease, despite evidence that a number of foods containing saturated fat, including dairy products such as yogurt, dark chocolate and unprocessed meat, are not associated with an increased risk of cardiovascular disease or diabetes [4,5,6,7]. The authors concluded that there was no solid evidence that the limits on saturated fat consumption, currently recommended by the U.S. guidelines, are able to prevent cardiovascular diseases or reduce mortality [2].

## 4. We Are Confronted with More Doubts Than Certainties: How Reliable Is Nutritional Epidemiology Research?

Ioannidis et al. [8] stated that the identified epidemiological associations of nutritional factors are often considered as causal effects, with the consequence to generate recommendations in public health guidelines. However, it is well known that associations do not mean causality. In recent meta-analyses of prospective cohort studies, nearly all foods revealed statistically significant associations with mortality risk [9,10]. Eating patterns are associated with many social and behavioral factors that vary over time, can affect health but cannot be fully taken into account as confounding factors in the analyses of available studies [11,12]. In other words, no current epidemiological cohort includes sufficient information that takes into account all the confounding factors typical of nutritional associations. Much of the literature assumes that the risk of disease is determined by the excess intake of certain components of food such as, for instance, carbohydrates or fats. However, chemical elements contained in food processing, contaminants and components that appear only in conditions or methods of food preparation (for example, cooking red meat) can affect the results. Further, nutrient combinations that confer potential cardiovascular risks may vary depending on the individual genetic background, metabolic profile, age and environmental exposure. Thus, untangling the potential influence of a single dietary component on a specific disease or mortality is difficult, if not impossible. Ioannidis concluded that readers and guideline developers generally ignore statements of causal inference and advocacy to public policy made by available nutritional epidemiology articles [8]. Observational studies, typical of nutritional epidemiology, are indeed unable to provide causal relationships between different food intake and risk of morbidity/mortality, they only provide evidence of associations [13]. Statements in guideline recommendations based only on observational studies should be avoided. For example, the US dietary guidelines, which recommend limiting the intake of saturated fatty acids to less than 10% of total calories in order to prevent the onset of cardiovascular disease, are not based on truly reliable scientific documentation [1].

## 5. From Uncertainties to Facts: What Are the Mechanisms Underlying the Metabolic Effects of Dietary Fats?

Complex and multifactorial are the possible nutritional or detrimental effects of dietary fats, because of the huge variety of lipid metabolites originating from either the enzymatic or non-enzymatic oxidation of polyunsaturated fatty acids (PUFA), cholesterol and phospholipids (Figure 2). Of note, enzymatically driven lipid oxidation is a biochemical process commonly regulated by negative feedback mechanisms and substrate availability, while the lipid autoxidation occurs randomly and can often overwhelm the physiological antioxidant system (Figure 2).

Focusing on PUFA oxidation, the enzymatic metabolism of omega-3 PUFA is well known to lead to products provided with anti-inflammatory and immunomodulatory properties [14]. Omega-3 competes with omega 6 PUFA as a substrate for cyclooxygenase and lipoxygenase, and in this way counteracts an excessive production of this second type of PUFA metabolites, some of them being strong pro-inflammatory mediators, thus potentially interfering with vascular homeostasis [14]. Recent clinical trial results suggest that increasing the intake of foods rich in omega-3 PUFA is beneficial for hypertension, also because of their metabolic products, namely omega-3 oxylipins, that reduce oxidative stress, protect the function of various membrane-related proteins and compete with omega-6 oxylipins in regulating vasodilator release [15].

Moreover, PUFA autoxidation products are easily generated by food heating, light exposure and/or improper storage, but at least the major omega-3 PUFA autoxidation product, namely 4-hydroxyhexenal, was shown to be much less cytotoxic than the corresponding omega-6 PUFA autoxidation product 4-hydroxynonenal (Figure 2) [16].

As for the omega-6 PUFA-derived 4-hydroxynonenal, a potential contribution to the pathogenesis of cardiovascular diseases appears to be given by oxidative derivatives of cholesterol (Figure 2), some of them clearly demonstrated to exert pro-atherogenic effects when present in excessive amounts [17].

## 6. From Observational Studies to Randomized Controlled Trials: What Is the Role of Omega 3 Fatty Acids in the Prevention of Cardiovascular Diseases?

An area of research that tested by means of controlled studies the benefit/risk profile of a dietary supplement is that of omega-3 fatty acids. Omega-3 are long-chain polyunsaturated fatty acids containing 18 to 22 carbon atoms and a varied number of double bonds, the first of which is in the n-3 position. Therefore, omega-3 fatty acids are also called n-3 polyunsaturated fatty acids (n-3 PUFA). They are essential fatty acids and thus they must be obtained from the diet. Products approved by regulatory authorities as drugs are an ethyl ester of polyunsaturated fatty acids with an eicosapentaenoic acid (EPA), and docosahexaenoic acid (DHA) content of not less than 85% and an EPA to DHA ratio of 0.9 to 1.5. On the basis of a number of observational studies [18,19,20], showing that an increasing number of fish servings per week was associated with a significantly lower rate of coronary mortality and sudden death, randomized clinical trials have been conducted in the primary and secondary prevention settings [21,22,23,24] and in patients with heart failure [25,26,27]. An exhaustive review of the effects of N-3 PUFA in the primary and secondary prevention of cardiovascular diseases was published in 2018 [28]. In the following sections the most recent trials will be discussed.

### 6.1. Primary Prevention

Two quite recent, randomized clinical trials tested n-3 PUFA in patients at high risk but without previous atherothrombotic events. The VITAL trial [21] randomized 25,871 patients in the USA (men ≥ 50 years and women ≥ 55 years) to n-3 PUFA (1 g daily) vs. placebo. Patients were followed up for a median period of 5.3 years. Primary end points were major cardiovascular events (a composite of myocardial infarction, stroke or death from cardiovascular causes) and invasive cancer of any type. Disappointingly, n-3 PUFA failed to lower the incidence of major cardiovascular events or cancer in comparison with the placebo.

The ASCEND trial [22] randomized 15,480 UK patients with diabetes mellitus to n-3 PUFA (1 g daily) vs. placebo. Mean follow-up was 7.4 years. The primary outcome was a first severe vascular event (i.e., nonfatal myocardial infarction or stroke, transient ischemic attack or vascular death, excluding intracranial hemorrhage). At the end of the study, among patients with diabetes but without evidence of cardiovascular disease, there was no statistically significant difference in the risk of serious vascular events between those allocated to receive n-3 PUFA and those who received a placebo. However, the analysis of the causes of death showed a 19% reduction in vascular deaths in the patients’ group supplemented with n-3 PUFA [18]. In other studies testing n-3 PUFA, there were no significant between-group differences in the rates of nonfatal serious adverse events [29,30]. Thus, regarding primary prevention, n-3 PUFA was unable to reduce the risk of cardiovascular events.

### 6.2. Secondary Prevention

In 1998 the GISSI Prevenzione [23] included patients who had survived a recent myocardial infarction and were randomly assigned to supplements of n-3 PUFA (1 g daily) or not. Over 3.5 years of follow-up, n-3 PUFA lowered the risk of all-cause death, non-fatal myocardial infarction and stroke by 10%. On the basis of this study, the European Regulatory Authority EMA approved PUFA for oral use in cardiovascular prevention after myocardial infarction in adults. However, this indication was subsequently withdrawn in 2018, due to the fact that additional studies conducted in patients exposed to updated secondary prevention treatments failed to confirm the beneficial effects of this strategy [31].

More recently the REDUCE-IT [24] trial enrolled 8179 patients with established cardiovascular disease (70% of the total population of the trial) or diabetes and other risk factors (the remaining 30%), with a fasting triglyceride level varying from 135 to 499 mg/dL. Patients were randomly assigned to receive 2 g of icosapent ethyl twice daily (total daily dose, 4 g) or mineral oil as a placebo. The primary endpoint was a composite of cardiovascular death, nonfatal myocardial infarction, nonfatal stroke, coronary revascularization or unstable angina. The key secondary endpoint was a composite of cardiovascular death, nonfatal myocardial infarction, or nonfatal stroke, that is, the classical MACE outcome of secondary prevention studies. Patients allocated to icosapent ethyl (4 g daily) showed a reduction in the primary and of key secondary endpoint by 25% and 26%, respectively. The beneficial effect was more evident in patients with a prior cardiovascular event. This study confirmed, 20 years after GISSI Prevenzione [23], the beneficial effects of omega 3 fatty acids in the secondary prevention setting, but with the use of a higher dose of purified icosopent ethyl. A higher dosage is likely to currently be necessary because contemporary patients are treated with more appropriate secondary prevention strategies, first of all the widespread use of statins. A confirmation that the effects of N-3 PUFAs may depend on their dosage and, as a consequence, on their circulating levels was provided by a subanalysis of REDUCE-IT (Bhatt DL. ACC/WCC 2020, Chicago, IL, USA). Looking at on-treatment EPA levels, the authors found significant associations with all measured cardiovascular outcomes. In other words, patients with highest EPA circulating levels showed the lowest rates of the different cardiovascular events, cardiovascular deaths and even total mortality.

On the basis of this study, EMA chose to reintroduce in 2021 the indication for the oral use of icosapent ethyl 4 g daily in the cardiovascular prevention strategies (https://www.ema.europa.eu/en/medicines/human/EPAR/vazkepa, accessed on 12 October 2022).

Recently, some criticisms have been raised on the REDUCE-IT trial results regarding the possibility that the comparator used in the study was a mineral oil with potential negative effects, mainly driven by proinflammatory effects [32], thus enhancing the potential favorable effect of icosapent ethyl in the trial. The issue of the placebo choice and the related negative changes in biomarkers was reviewed also by the US FDA, Health Canada, and the European Medicines Agency at the time of their process of approval. The conclusion was that any possible negative effect of mineral oil placebo on the 25% risk reduction seen in REDUCE-IT would have been quite limited and not sufficiently relevant to nullify the favorable effects of icosapent ethyl seen in the trial [33]. Another randomized clinical trial, STRENGTH [34], using high dosages of a different compound (carboxylic acid formulation of EPA and DHA) showed neutral results. Differences between the favorable results of REDUCED-IT and STRENGTH may be explained by the different pharmacological supplements tested in the studies and the different population of patients: more severe in REDUCE-IT than in STRENGTH. Patients with a documented previous atherothrombotic event accounted for more than 70% in REDUCE-IT while in STRENGTH the rate of patients in secondary prevention was approximately 56%. The difference in patient severity was documented also by the rate of MACE (CV death, non-fatal MI or stroke) that, over a similar period of follow-up, was approximately 13% in REDUCED-IT versus 8% in STRENGTH. Another possible reason for the difference is related to the circulating levels of EPA measured in the two trials. In front of similar baseline levels (26.1 µg/mL in REDUCED-IT versus 21.0 µg/mL in STRENGTH), the levels of EPA after 1 year of treatment were much higher in REDUCED-IT than in STRENGTH (144.0 µg/mL versus 89.6 µg/mL) [35].

### 6.3. Heart Failure

A potential pathophysiological rationale for using n-3 PUFA in heart failure has been described by a number of studies [26,27,36,37,38,39]. These findings, based upon intermediate endpoints, found a solid confirmation in the frame of a large scale randomized clinical trial, the GISSI-HF trial [25]. Patients with chronic heart failure of the New York Heart Association class II-IV, irrespective of etiology of heart failure and the level of left ventricular ejection fraction, were randomly assigned to n-3 PUFA 1 g daily (n = 3494) or placebo (n = 3481). The two co-primary endpoints were time to all-cause death, and time to all-cause death or admission to hospital for cardiovascular reasons. All-cause death occurred in 27% of patients allocated to the n-3 PUFA group and 29% in the placebo group (*p* = 0.041). Admission to hospital for cardiovascular reasons or death occurred in 57% of patients randomized to n-3 PUFA and in 59% of those in the placebo group (*p* = 0.009). In absolute terms, 56 patients needed to be treated for a median duration of 3.9 years to avoid one death or 44 to avoid one event such as death or admission to hospital for cardiovascular reasons. Thus, the beneficial effect was small albeit statistically significant, and occurred in the absence of serious adverse events.

### 6.4. The Issue of Atrial Fibrillation

Omega-3 fatty acids have been reported to have both benefits and risks with respect to incident atrial fibrillation [24,34,40] but large-scale, long-term randomized clinical trials with atrial fibrillation as the main endpoint are lacking. For this reason, patients who choose to take omega-3 fatty acids, especially in high doses, should be aware of the potential risk of atrial fibrillation and be followed up for the possible development of this common and potentially hazardous arrhythmia [41].

## 7. Final Considerations

Dietary habits have major implications as causes of death globally, particularly in terms of cardiovascular disease, cancer and diabetes. Precisely defining the role of a single component of diet in terms of cardiovascular risk is not an easy task, because nutritional epidemiology stands in a very slippery scientific context that makes it difficult, if not impossible, to take into account all the possible confounding factors. For this reason, and taking as an example the U.S. Dietary Guidelines that recommend a restriction of the intake of saturated dietary fat <10% of total calories, no recommendation can be considered as based on scientific evidence.

Concerning omega 3 fatty acids, when used as a drug they seem to maintain a therapeutic role only in the context of secondary cardiovascular prevention. Well conducted studies in primary prevention showed a neutral effect on the classical MACE, with just a weak signal of benefit on cardiovascular death reduction. In the secondary prevention setting, GISSI Prevenzione showed a favorable effect at low dosages (1 g per day), specifically on early reduction in sudden cardiac death. However, this result was obtained in the 90s, when other secondary prevention drugs (in particular statins) were practically non-existent in daily clinical practice. More recently, the REDUCE-IT trial showed very favorable effects of this drug at a high dosage (4 g/die) in a population of patients treated with up to date secondary prevention strategies. Despite some favorable evidence in patients with chronic heart failure, a treatment with n-3 PUFA in this clinical context is presently overlooked.

## Figures and Tables

**Figure 1 jcm-11-06652-f001:**
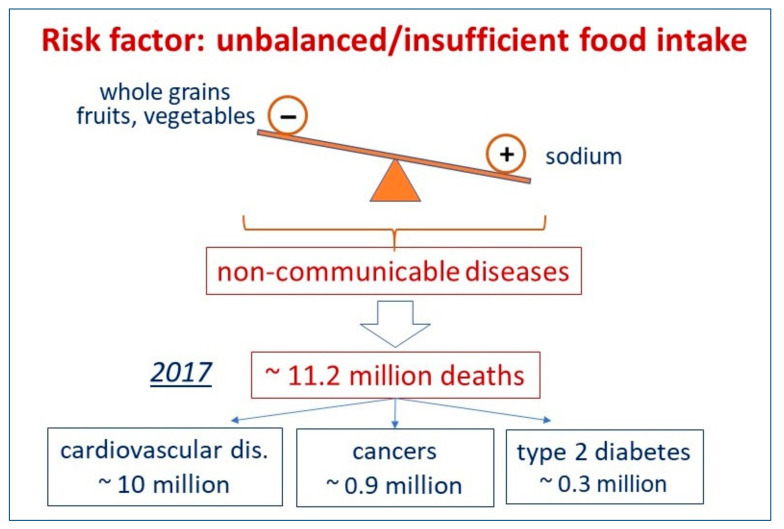
Unbalanced dietary factors, incidence/type of non-communicable diseases and deaths in 195 countries, in 2017 [1].

**Figure 2 jcm-11-06652-f002:**
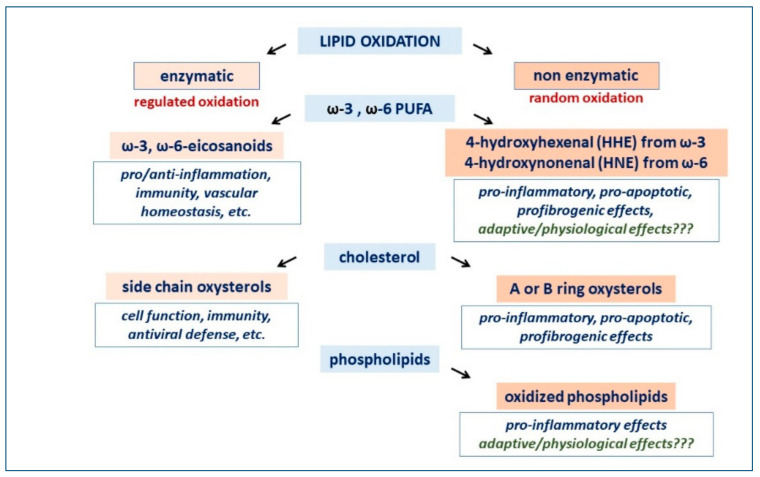
Enzymatic and non-enzymatic products of lipid oxidation.

## Data Availability

Not applicable.

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
