# Peer review of "Impact of Dietary Fats on Cardiovascular Disease with a Specific Focus on Omega-3 Fatty Acids"

_jcm, 2022, doi:10.3390/jcm11226652_

Round 1

Reviewer 1 Report

The authors provide a review article concerning the impact of dietary fats and omega 3 fatty acids on cardiovascular disease. This reviewer believes that the present manuscript is well written and easy to follow. Overall, the descriptions seem objective and neutral. However, in terms of a therapeutic role of omega-3 fatty acids in secondary cardiovascular prevention, additional considerations may be needed. As described by the authors, the REDUCE-IT trial was initially published as a positive study. However, the subsequent similar study, the STRENGTH, was a totally neutral trial [JAMA. 2020;324:2268-80]. Intense debates have been done recently, and some people believe that the lower incidence of MACE in the omega-3 fatty acid group in the REDUCE-IT was not driven by the benefit of omega-3 fatty acid but was driven by negative effect of the mineral oil comparator [Eur Heart J Suppl. 2020;22:J34-48, Eur Heart J. 2021;42:4807-17]. In fact, the recent sub-analysis of the REDUCE-IT showed proinflammatory effects of mineral oil [Circulation. 2022;146:372-9]. This reviewer believes that the topic is hot and should be included in the present paper.

Author Response

Many thanks to this reviewer for this relevant comment, raised again in a recent publication.

We added a sentence in the paragraph regarding the REDUCE-IT trial discussing this topic (Page 5, lines 47-52 and page 6 lines 1-4):

[…] Recently, some criticisms have been raised on the REDUCE-IT trial results regarding the possibility that the comparator used in the study was a mineral oil with potential negative effects, mainly driven by proinflammatory effects [31], thus enhancing the potential favorable effect of icosapent ethyl in the trial. The issue of the placebo choice and the related negative changes in biomarkers was reviewed also by the US FDA, Health Canada, and the European Medicines Agency at the time of their process of approval. The conclusion was that any possible negative effect of mineral oil placebo on the 25% risk reduction seen in REDUCE-IT would have been quite limited and not sufficiently relevant to nullify the favorable effects of icosapent ethyl seen in the trial. [32] […]

We added a sentence (page 6, lines 5-14) mentioning the neutral results of the STRENGTH trial but remembering the fact that a different compound was used and a different population of patients was recruited: less severe in STRENGTH than in REDUCED-IT.

Reviewer 2 Report

The authors described the current evidence of fat and PUFA nutrition in adults as a narrative review. The paper is readable and interesting. It would be wise to show the  a summary of the current evidence search with recommendation level.

Author Response

We thank the reviewer for this comment. However, our ambition was not to produce formal guideline recommendations with their level of evidence but just a summary on the effects of dietary fat and N-3 PUFA, easy readable for clinician.

Reviewer 3 Report

The article jcm-1995155 deals with an important issue such as the potential benefits associated with the intake of W3-fatty acids. Unfortunately, the authors do not help to clarify this issue; in fact, I think that increase the doubts. I am not sure whether the forgot to check all the trials and references against a positive effect of W3.but several neutral trials are missing

Several are the factors that hamper the impact of the article:

1.- They do not differentiate between natural W3 (fish intake) and the use of supplements. It is important to consider that not all supplements are equal.

2.- What about the differential effects of EPA vs DHA? I’d suggest mentioning the article by Martson NA et al Circulation 2019 140:1308.

3.- Similarly what about the Strength trial. It used same dose (4grs/day) that REDUCE-IT but with neutral results.

Author Response

Comments and Suggestions for Authors

The article jcm-1995155 deals with an important issue such as the potential benefits associated with the intake of W3-fatty acids. Unfortunately, the authors do not help to clarify this issue; in fact, I think that increase the doubts. I am not sure whether the forgot to check all the trials and references against a positive effect of W3.but several neutral trials are missing

We thank this reviewer for the comments.

Actually, neutral trials have been quoted regarding the studies conducted in primary prevention.

With respect to secondary prevention, we summarized just the results of trials that changed clinical practice following the approval of the regulatory authorities and the introduction of related recommendations in the guidelines (GISSI-Prevenzione in 1998 and REDUCE-IT more recently).

Several are the factors that hamper the impact of the article:

1.- They do not differentiate between natural W3 (fish intake) and the use of supplements. It is important to consider that not all supplements are equal.

In this article, we did not enter in the discussion regarding the potential effects of fish intake and pharmacological supplements of omega 3. Regarding dietary habits, we stressed the concept of the difficulties in interpreting data from nutritional observational research, stating that one the rare examples of randomized studies is that of trials testing omega-3 fatty acids. For this reason, just randomized clinical trials testing pharmacological compounds have been considered for this article.

2.- What about the differential effects of EPA vs DHA? I’d suggest mentioning the article by Martson NA et al Circulation 2019 140:1308.

3.- Similarly what about the Strength trial. It used same dose (4grs/day) that REDUCE-IT but with neutral results.

Several other trials and meta-analyses have been conducted in the last 20 years testing different compounds of N-3 PUFA with different (sometimes-conflicting) results. Considering just those using the same compound of REDUCE-IT: JELIS (Yokoyama M, Origasa H, Matsuzaki M et al.; Japan EPA lipid intervention study (JELIS) Investigators. Effects of eicosapentaenoic acid on major coronary events in hypercholesterolaemic patients (JELIS): a randomised open-label, blinded endpoint analysis. Lancet 2007;369:1090–1098) and CHERRY(Watanabe T, Ando K, Daidoji H et al.; CHERRY study investigators. A randomized controlled trial of eicosapentaenoic acid in patients with coronary heart disease on statins. J Cardiol 2017;70:537–544) produced favorable results.

Another trial (STRENGHT) conducted with a different compound (EPA+DHA) but a high dosages (4 gr daily) showed neutral results.

We added a sentence (page 6, lines 5-14) mentioning the neutral results of the STRENGTH trial but remembering the fact that a different compound was used and a different population of patients was recruited: less severe in STRENGTH than in REDUCED-IT.

Round 2

Reviewer 3 Report

The revised manuscript is a bit improved but  the most important concerns are still unanswered significantly reducing the article's impact

Author Response

Additional information has been added regarding the difference between the results of REDUCE-IT and STRENGTH as well as some considerations regarding EPA dosages, their circulating levels and related outcomes.